# A biocatalytic hydroxylation-enabled unified approach to C19-hydroxylated steroids

Junlin Wang[1,4], Yanan Zhang[2,4], Huanhuan Liu[2], Yong Shang[1], Linjun Zhou[2], Penglin Wei[3], Wen-Bing Yin[3], Zixin Deng[2], Xudong Qu [2] & Qianghui Zhou [1]

Steroidal C19-hydroxylation is pivotal to the synthesis of naturally occurring bioactive C19-OH steroids and 19-norsteroidal pharmaceuticals. However, realizing this transformation is proved to be challenging through either chemical or biological synthesis. Herein, we report a highly efficient method to synthesize 19-OH-cortexolone in 80% efficiency at the multi-gram scale. The obtained $C_{19}$-OH-cortexolone can be readily transformed to various synthetically useful intermediates including the industrially valuable 19-OH-androstenedione, which can serve as a basis for synthesis of C19-functionalized steroids as well as 19-nor steroidal drugs. Using this biocatalytic C19-hydroxylation method, the unified synthesis of six C19-hydroxylated pregnanes is achieved in just 4 to 9 steps. In addition, the structure of scler-osteroid B is revised on the basis of our synthesis.

[1] Sauvage Center for Molecular Sciences, Engineering Research Center of Organosilicon Compounds & Materials (Ministry of Education), Institute for Advanced Studies, College of Chemistry and Molecular Sciences, Wuhan University, Wuhan 430072, China. [2] Key Laboratory of Combinatorial Biosynthesis and Drug Discovery, Ministry of Education, School of Pharmaceutical Sciences, Wuhan University, Wuhan 430071, China. [3] State Key Laboratory of Mycology, Institute of Microbiology, Chinese Academy of Sciences, Beijing 100101, China. [4] These authors contributed equally: Junlin Wang, Yanan Zhang. Correspondence and requests for materials should be addressed to X.Q. (email: quxd@whu.edu.cn) or to Q.Z. (email: qhzhou@whu.edu.cn)

C19-hydroxylated steroids belong to a unique family of natural products with diverse bioactivities (Fig. 1a). For example, stereonsteroid A and B[1] exhibit excellent cytotoxicity against P-388 and HT-29 cancer cell lines. Besides the general cardiotonic activity, strophanthidol[2], 19-hydroxysarmentogenin[3–5] and ouabagenin[6–8] also possess antitumor activities. Bufadienolides bryophyllin A and C[9] show strong insecticidal activity against third instar larvae of silkworm (*Bombyx mori*) (LD$_{50}$ 3–5 µg g$^{-1}$). However, isolating these natural products from their sources is very low-yielding. For instance, bryophyllin A–C were isolated in 0.00018%, 0.00043%, and 0.00003% yields (w/w), repectively[10,11], which severely impedes further biological and pharmaceutical studies. Consequently, chemical synthesis is considered as an alternative approach for producing these steroidal compounds[4,5,7,8]. In recent years, selective C–H oxidation has emerged as one of the important reactions used in the chemical synthesis of steroidal compounds[12,13]. However, functionalization of the C19 methyl group in steroids through C–H oxidation still remains as a challenge. The synthetic strategies developed so far include the Norrish-type II photoreaction and oxidative fragmentation method[7,8] for steroids containing a C11 carbonyl group, and the oxygen radical-induced remote C–H hydroxylation under Barton's nitrite ester photocleavage[14,15] or Suárez-type conditions (Fig. 1b)[16–18]. Despite the effectiveness in C19-hydroxylation, all of these methods required prefunctionalization at specific positions of steroids, and necessitated strict reaction conditions and multi-step chemical transformations, often leading to low synthetic efficiency. Therefore, developing synthetic strategies for direct steroidal C19-hydroxylation is highly desirable.

Biocatalysis is emerging as a powerful and valuable tool for organic synthesis[19,20], and has the potential to be more efficient than chemical synthesis, particularly for functionalization of unactivated C–H bonds[21,22]. For example, P450 or other radical-mediated enzymes have been used to directly introduce functional groups to inert carbon moieties with high regio- and enantio-selectivity[20,23]. Combining biocatalytic C–H activation and chemical synthesis could enormously shorten the synthetic routes in natural product synthesis, as were demonstrated in a few of recently disclosed successful approaches towards some interesting natural products[24–30].

Inspired by these reported work, we start our exploration on using a biocatalytic approach to directly introduce the C19-hydroxyl group into steroids. It has been documented that a few filamentous fungi, belonging to the *Pellicularia* genus, able to hydroxylate the steroidal C$_{19}$–H bond[31]. It interests us that the strain and substrate pair, *Thanatephorus cucumeris* NBRC 6298 and cortexolone (**1**), are able to produce the C$_{19}$-OH cortexolone (**2**) (Fig. 1c) with a 20% conversion[31,32]. Albeit the low conversion, this method looks attractive to us owing to its easy operation and low cost, especially given that **2** could serve as a versatile intermediate to various important C19 functionalized steroids (Fig. 1c), and the currently available access of **2** has to go through a 13-step synthesis (Supplementary Fig. 1)[8].

Herein, we report our work on developing a steroidal C19 hydroxylation strategy, and a unified efficient synthesis of six bioactive pregnanes from **2** which examples the power of this strategy.

## Results

**Identification of the C19-hydroxylase**. We initially repeated the previously reported procedure of C19 bio-hydroxylation[31,32]. Consistent with the reports, the conversion of **1** to **2** in our hands was 20.3% (entry 1 in Table 1). Besides **2**, 11β-hydroxycortexolone (**3**, 12.5% conversion) was also obtained as well as

a number of minor side-products (trace III in Fig. 2). These side products were formed probably due to the non-specific activity of C19 hydroxylase or the existence of other enzymes. The responsible gene was cloned from the *T. cucumeris*, and Nine p450 and p450 reductase genes that robustly transcribed were identified through transcriptome analysis (Supplementary Table 1). These p450 genes were cloned under the strong promoter *PamyB*, and individually transformed into the model filament fungus *Aspergillus oryzae* NSAR1[33] for evaluation of the activity. To our delight, *tcP450-1*, one of the p450 genes, is confirmed to be able to convert **1** to **2**. However, like *T. cucumeris*, this recombinant strain also produced **3** and other minor side products (Fig. 2, trace II), suggesting that TcP450-1 has a relaxed specificity and generates all observed cortexolone derivatives. Based on the ratio of each product, the maximum conversion of **2** by TcP450-1 is inferred to be 48% (calculated by dividing its peak area into the sum of all products' peak area), significantly higher than the current value of 20.3%, and indicating a potential for further improvement.

**Optimization of the C19 hydroxylation activity**. In spite of easy manipulation, basal activity in *A. oryzae* is also able to convert **1** into a reduced product, C$_{20}$–OH cortexolone (**4**) (Fig. 2, trace I and II), which results in very low conversion of the C$_{19}$–OH product. During this study, Lu et al also identified the C$_{19}$–OH hydroxylase and overexpressed it in *Pichia pastoris*[34]. Although no reduced products were generated, the methylotrophic yeast system still results in very low conversion. As the usefulness of both heterologous systems are limited, we, therefore, go back to optimize the original *T. cucumeris* system. We then worked on the optimization of fermentation conditions. As **1** was found partially precipitated in the fermentation broth, we reduced the concentration **1** from 0.5 g L$^{-1}$ to 0.25 g L$^{-1}$ which, as a result, increased the conversion of **1** to 27.0% (entry 2 in Table 1 and Supplementary Fig. 2). Subsequent using liquid media to develop a seed culture could continue to improve the conversion to 29.0% (entry 3 in Table 1 and Supplementary Fig. 2).

As TcP450-1 is an ion-dependent enzyme, and metal ions could play a significant role in cell growth and enzyme activity, we evaluated the effect of Mg$^{2+}$, Zn$^{2+}$, Mn$^{2+}$, and Fe$^{2+}$ (each 1 mM L$^{-1}$) on the substrate conversion. It was found that Fe$^{2+}$ was the best among the ions evaluated: the conversion could be increased to 34.9% and 45.1% when the concentration of Fe$^{2+}$ added into the fermentation culture was 1.0 mM L$^{-1}$ and 1.5 mM L$^{-1}$, respectively (entries 4 and 5 in Table 1 and Supplementary Figs. 3 and 4).

As the obtained highest conversion is close to the theoretic value (48%), we shifted the focus onto optimization of the enzymatic reaction. Although protein engineering is generally an effective way to improve an enzymes' substrate specificity[20,22], this direction could be very challenging for TcCYP450-1 due to its membrane-binding properties and lack of reliable homologous structural models. Therefore, instead of changing the enzyme we chose to modify the substrate. It was described that progesterone, a steroid structurally similar to cortexolone (**1**), was not able to be hydroxylated in C19[31], which implied that the hydroxyl substitution at C17 or C21 in **1** could be crucial for the enzyme's specificity. Therefore, engineering these hydroxyl groups might be helpful. To test this assumption, 21-acetyl-cortexolone (**5**) and 17-acetyl-cortexolone (**6**)[35] are employed (Fig. 2). Feeding of **5** to the broth culture resulted in an identical product profile to that of **1** (trace V in Fig. 2). To our delight, when **6** was used as the substrate, the formation of 11β-OH and other minor side products was greatly reduced, and **2** was formd in a significantly high conversion (80%) (entry 6 in Table 1; trace VI in Fig. 2).

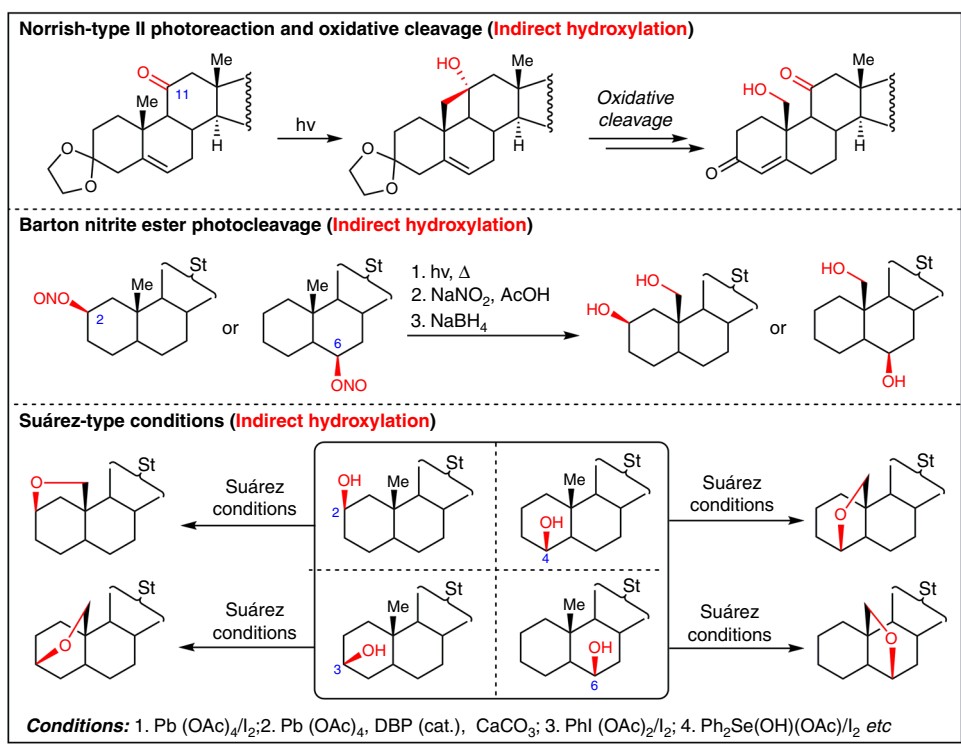

**Fig. 1** Chemical and biocatalytic approache to access 19-Hydroxylated steroids. **a** Representative bioactive 19-hydroxylated steroids. **b** Chemical approaches to C19 hydroxylation of steroids. **c** Enzymatic C19 hydroxylation of steroids

**Table 1 Fermentation Conditions Optimization**

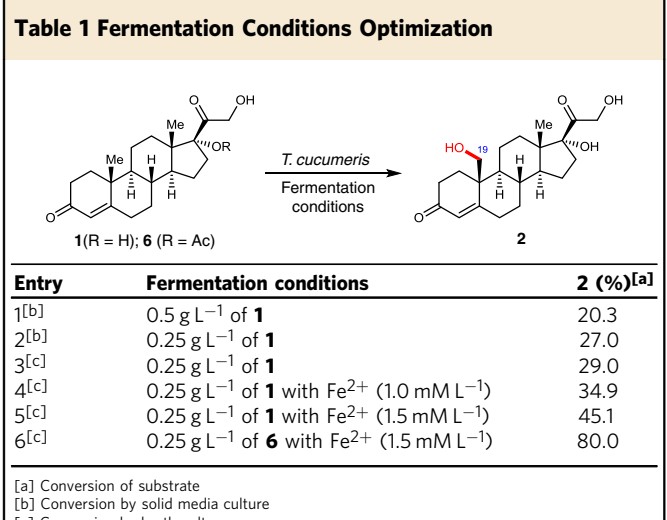

| Entry | Fermentation conditions | 2 (%)[a] |
|---|---|---|
| 1[b] | 0.5 g L$^{-1}$ of 1 | 20.3 |
| 2[b] | 0.25 g L$^{-1}$ of 1 | 27.0 |
| 3[c] | 0.25 g L$^{-1}$ of 1 | 29.0 |
| 4[c] | 0.25 g L$^{-1}$ of 1 with Fe$^{2+}$ (1.0 mM L$^{-1}$) | 34.9 |
| 5[c] | 0.25 g L$^{-1}$ of 1 with Fe$^{2+}$ (1.5 mM L$^{-1}$) | 45.1 |
| 6[c] | 0.25 g L$^{-1}$ of 6 with Fe$^{2+}$ (1.5 mM L$^{-1}$) | 80.0 |

[a] Conversion of substrate
[b] Conversion by solid media culture
[c] Conversion by broth culture

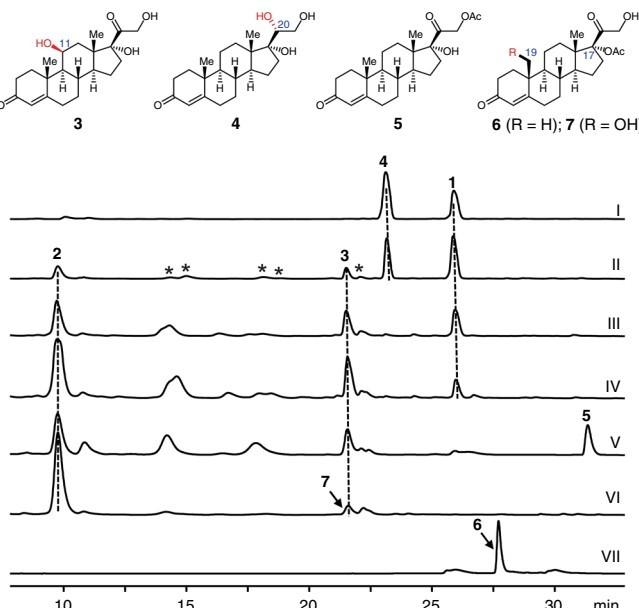

**Fig. 2** Production of hydroxylated cortexolones using *A. oryzae-tcP450-1* and *T. cucumeris*. (I) *A. oryzae* integrated with empty vector pTAex3, fed with 1; (II) *A. oryzae-tcP450-1*, fed with 1; (III) *T. cucumeris* under the conditions described[31, 32], fed with 1; (IV) *T. cucumeris* with the optimized condition, fed with 1; (V) *T. cucumeris*, fed with 5; (VI) *T. cucumeris*, fed with 6; (VII) standard 6. Samples were taken and analyzed at the fifth day (for *A. oryzae*) and third day (for *T. cucumeris*) after substrates were fed. Peaks with asterisk are other minor hydroxylated cortexolones

Furthermore, the key intermediate C$_{19}$–OH 17-acetyl-cortexolone (7) was identified in the early stage of biocatalysis (trace VI in Fig. 2 and Supplementary Fig. 5). Hence, the C17 acetyl group in 6 played a role in not only the C19 hydroxylation specificity, but also the C19 hydroxylation activity as 6 was able to be fully converted into 2 within 24 h (Supplementary Fig. 5). Thus, we postulated that 6 is quickly hydroxylated into 7 which is then gradually hydrolyzed into 2 regioselectively, and the difference in hydroxylation and hydrolysis reactivities ensures the high C19-specficity. The high efficiency of this biocatalytic approach is obvious considering that the precedent chemical approach requires 13 steps from cortisone acetate and suffers from a very

low overall yield (2%) (Supplementary Fig. 1)[8]. Following this method, we scale up the biotransformation and obtain over five grams of 2.

**Synthetic versatility of C$_{19}$–OH cortexolone (2).** Armed with this scalable and robust C19 hydroxylation method, we started to examine its versatility by transforming 2 into other synthetically useful intermediates (Fig. 3). The first exampled transformation was the treatment of 2 with basic hydrogen peroxide[36]. We were delighted to find that the selective β-face epoxidation of 2 took place to provide epoxide 8 as the sole diastereomer (58% yield) whose formation, we believe, was stereo-specifically directed by the C19 hydroxy group in 2. When 2 was subjected to Pd/C mediated hydrogenation conditions, the A/B cis-configurated product 9 was formed with moderate diastereoselectivity. DDQ-mediated desaturation of the A ring in the protected form of 2 with its 19,21-diols protected by acetyl group[37], intermediate 10 was obtained with 81% yield over two steps from 2. The allylic oxidation of C6 in 2 was realized through a two-step sequence involving the formation of enol acetyl ester and PhIO-mediated oxidation[38], resulting in a β-allylic alcohol 11 (overall yield of 52% from 2). Additionally, the TMSI-promoted[39] selective C17 dehydroxylation of 2 resulted in the highly desired 19-hydroxydeoxycorticosterone (19-OH-DOC) 12[40] in 84% yield (β:α = 20:1). This transformation is intriguing in terms of its high chemo- and diastereo-selectivity since the C19,21 diols as well as C3,20 carbonyl groups remained untouched during the process. Literature precedents[41,42] showed that the α-hydroxyl ketone motif in the side chain of 2 and 12 could act as a functional handle for direct appendage of butenolide motif (13), which is ubiquitous in cardiotonic steroids. As another important exampled application, 2 was transformed to 19-hydroxy-androstenedione 14 in 84% yield through a facile oxidative cleavage of the side chain. Both 12 and 14 are key intermediates in pharmaceutical industry as were used in the preparation of more than ten 19-nor steroidal drugs such as 19-nor-DOC[40], norethisterone, and ethinyloestradiol (Supplementary Figs. 6–8). It is worth noting that the preparation of either 12 or 14 has to be through more than 8 steps using the current conventional chemical methods (see Supplementary Figs. 6, 9–11 for details).

**Collective synthesis of C$_{19}$-OH pregnanes.** To further demonstrate the usefulness of this strategy, we have accomplished the unified synthesis of five bioactive pregnanes in stereonsteroid family, with intermediate 12 as the starting point (Fig. 4). Our embarking on these biologically important stereosteroids started from constructing the trans A/B ring junction. Considering the previously incorporated C19 hydroxy group would take a deleterious directing effect, as demonstrated in the preparation of 9, the two hydroxy groups on C19 and C21 were protected as TBS ethers to provide 15. Then, a one-pot reaction involving Pd/C-catalyzed hydrogenation of C=C double bond and subsequent NaBH$_4$-mediated reduction of 3,20-diketones was conducted to deliver 16 which was a key intermediate since it already possessed the desired trans-fused A/B ring system. Removal of the TBS protecting group from 16, followed by NaIO$_4$-mediated oxidative cleavage of vicinal diol in the side chain lead to aldehyde 17 in excellent yield (95% over two steps). After a Wittig olefination was performed in 93% yield, the synthesis stereonsteroid A (18) was achieved in just five steps from 12. Then, three natural pregnanes, sclerosteroid A (20), ceratosteroid C and D (21 and 19), were concurrently obtained through selective acetylation of 18 using acetic anhydride. Moreover, ceratosteroid C (21) was further transformed into stereonsteroid B (22) quantitatively through a facile Dess-Martin periodate oxidation. These five

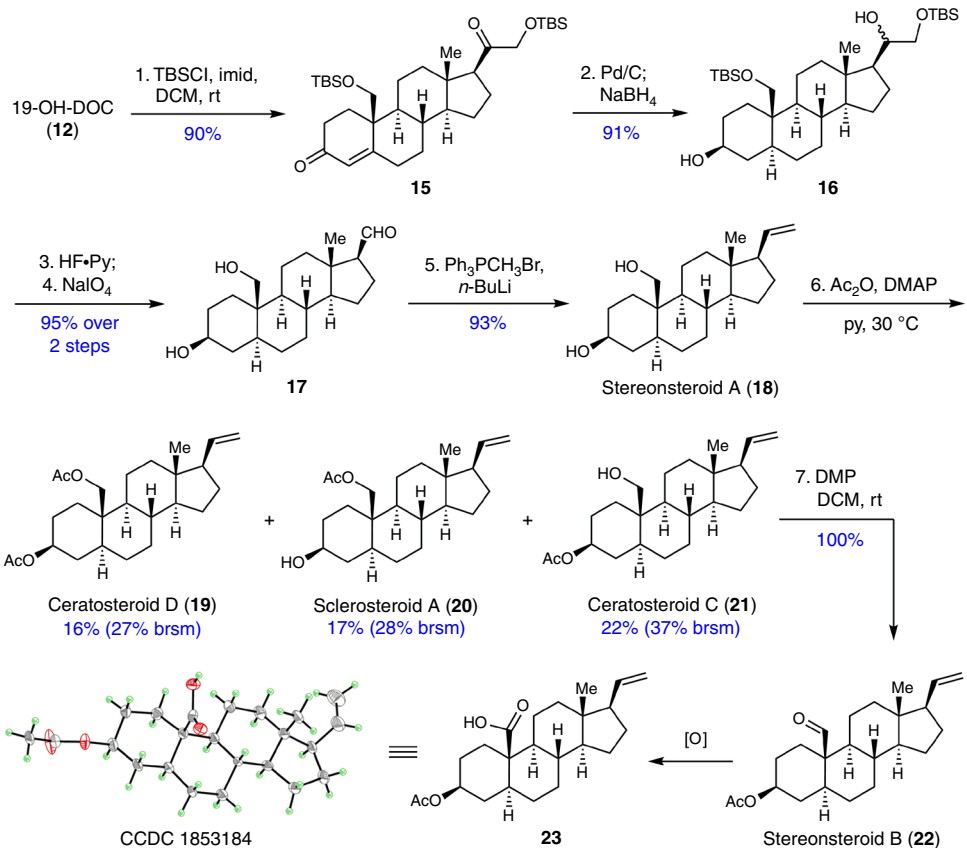

**Fig. 3** Synthetic transformations of 19-OH-cortexolone. 19-OH-cortexolone was readily transformed into various synthetically useful steroidal intermediates through epoxidation, hydrogenation, desaturation, allylic oxidation, cleavage of the C17-hydroxy group or the side chain

**Fig. 4** Unified synthesis of the stereonsteroid family. Five stereonsteroid family members including sclerosteroid A, stereonsteroid A and B, ceratosteroid C and D were synthesized from the common intermediate 19-OH-DOC in 5 to 7 steps

**Fig. 5** Structure revision of sclerosteroid B based on synthesis. Both the proposed structure and corrected structure of sclerosteroid B were synthesized from 19-OH-DOC. In addition, their anticancer activities were evaluated

synthetic samples (**19** to **22**) are spectroscopically consistent with those reported in the literature[1,43,44]. Additionally, their structures were unambiguously confirmed by X-ray crystallographic analysis of **23** (CCDC 1853184) which is a carboxylic acid derivative of **22**.

Encouraged by the successful step-economic synthesis of the five stereonsteroid compounds, we shifted our efforts to the synthesis of sclerosteroid B which was reported to possess promising anti-inflammatory activity[43]. As depicted in Fig. 5, the synthesis also started from **12**, which was transformed to the conjugated enol acetyl ester **24** through concurrent acetylation of the C19, C21 diol in an excellent yield. Then, one-pot procedure involving NaBH4-mediated reduction and subsequent NaIO4-mediated oxidative cleavage of the in situ formed vicinal diols in the side chain to provide the aldehyde **25** in 75% yield[45]. It is worth mentioning that, through this one-pot process, three transformations had been achieved, including diastereoselective reduction of the C3 carbonyl, reduction of the C20 carbonyl, as well as removal of the acetyl protection of the C19,21 diol. Finally, **25** was transformed to the reported structure of sclerosteroid B (**26**) through a one-pot Wittig olefination and bis-acetylation. However, we found that the spectra of the synthetic sample **26** do not match with what was reported[43]. After careful examination of the spectral data, we surmised that the internal double bond might be assigned incorrectly in the reported structure, and C4-C5 position might be the correct location for the double bond. To confirm our hypothesis, we started our synthesis of the proposed molecule in just two steps from **12** by following a similar strategy. After **12** was transformed to aldehyde **27** through a one-pot process of Luche reduction[46] and NaIO4-mediated oxidative cleavage of the side chain, another one-pot operation involving Wittig olefination and acetylation of C19,21 diol transformed **27** into the revised sclerosteroid B (**28**) in 78% yield. Gratifyingly, the NMR spectra of the synthetic sample **28** is consistent with those reported. Thus, we concluded that **28** is the correct structure of sclerosteroid B whose synthesis was achieved in just two steps from **12**. In the biological assay, both **28** and **26** showed cytotoxicity activity against human colon cancer cell line HT29 by inducing cell apoptosis (IC50 = 59.76, and 169.50 μM for **28** and **26**, respectively) (see Supplementary Fig. 12). These results

suggest that the location of the double bond is important for the anticancer activity.

## Discussion

C19 hydroxylation is a unique feature of some natural-occurring bioactive steroids, but these 19-hydroxylated steroids are normally in low abundance in nature, which has severely impeded their biological and medicinal investigations. Worse still, this modification is very challenging from both the chemical and biological synthesis perspective. Traditional chemical approaches for steroidal C19 hydroxylation require prefunctionalization at specific positions and multi-step chemical transformations (see Fig. 1)[7,8,14–18]; necessitate harsh reaction conditions and suffer from low synthetic efficiency. To the best of our knowledge, only one example of biocatalytic direct C19 hydroxylation has been reported so far (from **1** to **2**)[31], but the synthesis suffered from very low efficiency (20% conversion).

To provide good solutions to this synthetic challenge, we turned our focus on developing an efficient and broadly applicable approach to access the 19-hydroxylated steroids by combining the power of biocatalysis and synthetic chemistry. Initially, we cloned the C19-hydroxylase TcP450-1. Through heterologous expression in the *A. oryzae* NSAR1, we conjectured that its low activity could be majorly stemmed from poor reaction specificity (generating other hydroxylated products). However, protein engineering to alter its substrate specificity could be a very difficult task since this protein has two transmembrane segments (Supplementary Fig. 13), and no similar proteins could be used in homologous structural modeling. To address this problem, we employed an unusual strategy—manipulating the substrate instead of engineering the enzyme. Fortunately, we identified that acetylation of the C17-OH of cortexolone (**1**) could significantly improve the enzyme's specificity and efficiency (conversion from 20.3 to 80%), allowing C19–OH-cortexolone (**2**) be formed in a scalable way.

The versatile C19–OH-cortexolone (**2**) could be transformed into an array of synthetically useful intermediates (**8-14**), among which 19-hydroxydeoxycorticosterone (19-OH-DOC, **12**) and 19-OH-androstenedione (**14**) are particularly attractive. Both of

them were prepared from **2** in just one step in excellent yields. In contrast, these two compounds are currently made in 8 to 11 steps by conventional chemical methods with low efficiency, and nasty reagents are required as well[40,47,48] (Supplementary Figs. 6, 9–11).

**12** retains the 17-β configured stereocenter which is ubiquitous in bioactive steroids. Moreover, the C17 α-hydroxy ketone side chain of **12** provides very useful handles for further structural manipulations. For instance, by reacting with Bestmann ylide, this hydroxymethyl ketone moiety can be efficiently converted into the γ-crotonolactone[41,42], the core structure of the large cardenolide family. As this work afforded a very convenient approach to access **12**, a general synthetic strategy was conceived for preparation of C19-OH 17-β steroids with using **12** as the key precursor. The effectiveness of this strategy was well-validated through our achievement of the unified synthesis of six C19-hydroxylated pregnanes as well as the synthesis and structural revision of sclerosteroid B. The primary C19-OH precursor **2** has already been commercialized through our collaboration with Adamas, a chemical reagent company.

Because the C19 hydroxymethyl group in **14** can be readily removed through the facile oxidative decarboxylation, **14** has become the highly valuable common intermediate to manufacture the 19-nor steroidal drugs in industry. Currently, more than ten 19-nor steroidal drugs are made employing **14** as the key intermediate (Supplementary Figs. 7, 8). Nevertheless, current process for the preparation of **14** is very lengthy (8 steps) and suffers from low efficiency and toxic reagents (Supplementary Figs 10-11 for details). In stark contrast, by using our method, **14** could be afforded in just one step from 19-OH-cortexolone (**2**) or 3 to 4 steps from cortexolone 21-acetate (RSA) (Supplementary Fig. 14). It is worthwhile mentioning that RSA is an inexpensive precursor for making hydrocortisone and produced in bulk in the steroid industry (the current price is less than 100 USD kg$^{-1}$). Furthermore, considering the reagents and reaction conditions employed, it is evident that our method is much greener and more economical than the current chemical synthesis, so it has the potential to replace the manufacturing process of **14**.

In addition to the above above-mentioned work, we predict the protection group in 17-OH could lay a solid foundation for further expanding the substrate scope of C19 bio-hydroxylation. Exploration of the relationship between the enzymes' specificity and C17-OH protection groups as well as other steroid variants with 17-acetyl are ongoing in our laboratory and will be reported in due course.

## Methods

**Transcriptome analysis of *T. cucumeris*.** *T. cucumeris* was inoculated into the YD liquid medium and cultured at 30 °C, 200 rpm for 2 d before transferred into a 250 mL Erlenmeyer flask containing 100 mL of YD liquid medium (5% (v/v)). Cortexolone (50.0 mg, final concentration 0.5 g L$^{-1}$) is dissolved in DMF (0.5 mL) and filtered for sterilization, then added to the medium. The fermentation of the culture is then continued under the same conditions until a small amount of product is detected by LC-MS. The mycelium was centrifuged down and immediately stored in liquid nitrogen. The total RNA was extracted for transcriptome sequencing at Genewiz (Suzhou, China).

**Heterologous Expression of TcP450-1 in *Aspergillus oryzae*.** The gene of *tcP450-1*(P450$_{19OH}$) obtained by PCR amplification of the cDNA of *T. cucumeris* using primer *tcP450-1*-F/R (Supplementary Table 2) and cloned into pTAex3 at the *Eco*RI site to yield the expression plasmid pWHU2490. *A. oryzae* NSAR1 was grown on PDA for 5 d. The seed culture was inoculated into a 250 mL Erlenmeyer flask containing 50 mL of DPY medium. After incubation at 30 °C for 2 d, mycelia were collected and washed twice with an aqueous solution of NaCl (0.9% w/w). Protoplast was prepared by using Yatalase (Takara; 3.0 mg mL$^{-1}$) and Lysing Enzymes (Sigma; 4.5 mg mL$^{-1}$) in Osmotic Medium (0.6 M of MgSO$_4$, 10 mM of Na$_2$HPO$_4$, pH 5.8) at 30 °C for 3 h. The Osmotic Medium containing Protoplasts was filtered and centrifuged at 1157 × *g* for 5 min, then washed with trapping buffer (0.6 M D-Sorbitol, 0.1 M Tris-HCl, pH 7.0) and STC buffer (1.2 M of D-Sorbitol,

10 mM of CaCl$_2$, 10 mM of Tris-HCl, pH 7.5). Then, the protoplasts were diluted by STC buffer till 10$^8$–10$^9$ cells mL$^{-1}$, and used for plasmid transformation immediately[33,49,50].

Plasmids pWHU2490 and pTAex3 were added into the protoplast in order to generate the corresponding TcP450-1 expression strain and control strain, respectively. After the protoplast was incubated on ice for 60 min, 1 mL PEG4000 solution (250 g L$^{-1}$ PEG4000, 100 mM CaCl$_2$, 600 mM KCl, 50 mM Tris-HCl pH 7.5) was added. After 30 min incubation at room temperature, STC buffer (4 mL) solution was added. The transformation system was then poured onto the Czapek-Dox agar plate supplemented with appropriate nutrients, and incubated at 30 °C for 3–7 d. Transformants were picked out and streaked onto the corresponding auxotrophic Czapek-Dox plates for growth for a few days. The recombinant strains (mWHU2487 and mWHU2488) were then transferred to the corresponding auxotrophic Czapek-Dox liquid medium, and shaken at 30 °C for 2 d to extract DNA. The genotype of the transformants mWHU2487 and mWHU2488 were verified by PCR using primers Pamy-gt-F and Tamy-gt-R (Supplementary Table 2, Supplementary Fig. 15).

**Biotransformation with *T. cucumeris*.** Using liquid media to develop seed culture: *T. cucumeris* was inoculated into the YD liquid medium and cultured at 30 °C at 200 rpm for 2 d[31,32]. Then, 5 mL seed culture was transferred to a 250 mL Erlenmeyer flask containing 100 mL of YD liquid medium, and incubation was continued under the same culture conditions for 2 d. Using solid media to develop seed culture: *T. cucumeris* was grown on PDA for 7 d, crushed and inoculated into a 250 mL Erlenmeyer flask containing 100 mL of YD liquid medium, and then further cultured at 30 °C at 200 rpm for 2 d. Cortexolone (50.0 mg, final concentration 0.25 g L$^{-1}$ or 0.5 g L$^{-1}$) was dissolved in DMF (0.5 mL) and sterilized by filtration, and added into the culture broth. After the culture was continued to be fermented under the same conditions for 3 d, it was subjected to EtOAc extraction (3 × 100 mL EtOAc for each 100 mL of the culture, sonicated for 15 min) The combined organic extracts were dried under reduced pressure to result in metabolites which were subsequently re-dissolved with MeOH (2 mL) and filtered through a 0.22 μm membrane filter to remove particles. HPLC analysis was performed at a flow rate of 1 mL min$^{-1}$ over a 40 min gradient program: T = 0 min, 25% B; T = 10 min, 25% B; T = 25 min, 70% B; T = 27 min, 25%; T = 40 min, 25% B (A = H$_2$O, B = CH$_3$CN) or a 50 min gradient program: T = 0 min, 20% B; T = 20 min, 20% B; T = 40 min, 70% B; T = 45 min, 20% B; T = 50 min, 20% B (A = H$_2$O, B = CH$_3$CN).

**Biotransformation with recombinant *A. oryzae* strain.** Recombinant *A. oryzae* strains are inoculated into a 250 mL Erlenmeyer flask containing 100 mL MPY medium (maltose-polypeptone-yeast extract: 3% maltose, 1% polypeptone, and 0.5% yeast extract)[49]. The strains were cultured at 30 °C and 200 rpm for 2 d. Cortexolone (50.0 mg) was dissolved in DMF (0.5 mL, final concentration 0.25 g L$^{-1}$), were sterilized by filtration, then added to the medium. The culture was continually incubated under the same culturing conditions for 5–10 d, and then subjected to EtOAc extraction (3 × 100 mL EtOAc for each 100 mL of the culture, sonicated for 15 min) The combined organic extracts were dried under reduced pressure to result in metabolites which were subsequently re-dissolved with MeOH (2 mL) and filtered through a 0.22 μm membrane filter to remove particles before HPLC analysis.

**Biotransformation with different metal ions.** *T. cucumeris* was into the YD liquid medium and cultured at 30 °C and 200 rpm for 2 d. Then, 5 mL seed culture was is transferred into a 250 mL Erlenmeyer flask containing 100 mL of YD liquid medium, and the incubation was continued under the same culture conditions for 2 d. A solution of cortexolone (25.0 mg in 0.5 mL of DMF, final concentration 0.25 g L$^{-1}$) and a solution of MgSO$_4$, ZnSO$_4$, MnCl$_2$ or FeSO$_4$ (final concentration 1.0 mM L$^{-1}$) were sterilized by filtration, and added into the medium. After the culture was fermented under the same conditions for 3 d, it was subjected to EtOAc extraction (3 × 100 mL EtOAc for each 100 mL of the culture, sonicated for 15 min). The combined organic extracts were dried under reduced pressure. Metabolites were subsequently re-dissolved with MeOH (2 mL) and filtered through a 0.22 μm membrane filter to remove particles before HPLC analysis.

**Biotransformation using the optimized condition.** *T. cucumeris* was inoculated into the YD liquid medium and cultured at 30 °C and 200 rpm for 2 d. Then, 5 mL seed culture was transferred into a 250 mL Erlenmeyer flask containing 100 mL of YD liquid medium, and incubation was continued under the same culture conditions for 2 d. A solution of substrate **1**, **5** or **6** (25.0 mg dissolved in 0.5 mL of DMF, final concentration 0.25 g L$^{-1}$) and an aqueous solution of FeSO$_4$ (1.5 mmol L$^{-1}$) are filtered for sterilization, were sterilized by filtration, and added into the medium, and fermented under the same conditions for 3 d, and then subjected to EtOAc extraction (3 × 100 mL EtOAc for each 100 mL of the culture, sonicated for 15 min). The combined organic extracts were dried under reduced pressure to result in metabolites which were subsequently re-dissolved with MeOH (2 mL) and filtered through a 0.22 μm membrane filter to remove particles before HPLC analysis.

**Reporting summary**. Further information on research design is available in the Nature Research Reporting Summary linked to this article.

## Data availability
The sequence of the P450 genes reported in this paper has been deposited in GenBank under accession number MK309346-MK309355. The X-ray crystallographic coordinate for structure of **23** reported in this study has been deposited at the Cambridge Crystallographic Data Centre (CCDC) (number 1853184), and can be obtained free of charge from CCDC via www.ccdc.cam.ac.uk/data_request/cif. The raw data underlying Supplementary Figs. 2–4, 12 and 15 are provided as a Source Data file. All other relevant data are available from the corresponding authors.

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

## Acknowledgements

We are grateful to the National Key R&D Program of China (2018YFC1706200), the National Natural Science Foundation of China (Grant Nos. 21602161, 21871213, 21801193), and the China Postdoctoral Science Foundation (Nos. 2016M602339, 2018M642894) for financial support. We thank Prof. Ikuro Abe for gifting the *A. oryzae* NSAR1, Dr. Xianggao Meng (CCNU) and Ms. Wei Yan (WHU) for X-ray analysis, Mr Peng Wang and Dr. Chevula Gurumurthy (WHU) for the preparation of some substrates, Dr. Han-Qing Dong (Arvinas, Inc.) and Dr. Yahu Liu (GNF, Inc.) for assistance with the preparation of the manuscript.

## Author contributions

X.Q., Q.Z. and Z.D. conceived this project; J.W., Y.Z., H. L., L.Z., Y.S., P.W. and W.Y. performed the experiment; X.Q., Q.Z., J.W. and Y.Z. analyzed data; and Q.Z., X.Q., J.W. and Y.Z. wrote the manuscript.

## Additional information

**Competing interests:** The authors declare no competing interests.

