## [Peer Review File · Nature Communications]

Reviewers' comments:

Reviewer #1 (Remarks to the Author):

The manuscript entitled "A Biocatalytic Hydroxylation Enabled Unified Approach to C19-hydroxylated Steroids" by Wang et al describes the identification of one of the 9 transcribed p450s from *Thanatephorus cucumeris* NBRC 6298 as a C19 hydrolyase for cortexolone. Optimization of their whole-cell bio-catalysis approach demonstrated that the wt strain was superior to heterologous expression and that lower substrate loading supplemented with Fe(II) enabled 45% conversion of cortexolone into the desired C19 hydroxylated steroid core. Rather than engineering the P450 to decrease unwanted C11 hydroxylation, the authors screened two modified substrates to identify 17-acetyl-corexolone as a substrate that suppresses C11 hydroxylation, providing nearly exclusively C19 hydroxylated product. Using well established chemistry from the vast steroid synthesis literature the authors show a variety of transformation of the C19 hydroxylated substrate, including formation of C19-hydroxy-androstenedione a key intermediate in the formation of multiple steroid-based medicines. They also show the conversion of the C19 hydroxylated intermediate into six known steroid natural products. This includes sclerosteroid B, which they correct the structural missassignment present in the literature.

A notable achievement of this study is the high yielding whole-cell biocatalysis step providing C19 hydroxylation. This clearly fills a need in steroid chemistry. C19 hydroxylation is important as evidenced by the utility of C19-hydroxy-androstenedione, however it is somewhat of an unsolved problem in the literature. While Baran's recent work with ouabagenin demonstrates a succinct route to C-19, C11 oxidized steroid cores, no effective direct route to C19 hydroxyl C11 saturated steroids currently exists. As such this work is an important addition to the literature.

The coupling of biocatalysis with synthetic chemistry to access complex natural products has progressed significantly over the past few years, with a number of notable examples. This work compared favorably to these recent studies. For example, the pioneering work of Arnold and Stoltz in the use of late-stage hydroxylation to access the natural product nigelladine A stands out as useful comparator. Like this study a late-stage P450 oxidation is used to access a hydroxylated intermediate that is ultimately converted into the natural product. In the current study, significantly better conversion is obtained than in the Arnold/Stoltz example. This has a major impact on the efficiency of the work. In addition the current study access a hydroxylated intermediate that has multiple, immediate applications (as demonstrated by their synthesis of 6 natural products and a precursor to 10 medicinal steroids), where as the Arnold/Stoltz example generated a single natural product. Biocatalytic routes to jorunnamycin A and saframycin A by Oikawa and Oguri also provide a late stage transformation that enable the synthesis of multiple natural products. This example is powerful as it demonstrates a significant increase in complexity catalyzed by the biocatalyst, however it suffers from low yield (20%), an extremely unstable and challenging to scale enzyme, and relies on an enzyme from the saframycin A biosynthetic pathway to make saframycin – thus it is not surprising that it works. Lastly the work by Renata on early stage oxidation demonstrates the power of construction of oxidized building blocks by P450 mediated-chemistry, though its use in the early stage of the synthesis is less impactful since it does not take advantage of the selectivity imparted via biocatalysis to effect region-selective chemistry in highly functionalized molecules. As such this current example is perhaps one of the best examples in the literature of complex biocatalysis in natural product synthesis.

The reassignment of the structure of sclerosteroid B is a pleasant addition to the study, though is of marginal impact. The original, and only report of sclerosteroid B provides no experimental support for the 5,6 ene position and further sclerosteroids isolated by the same group possessed a 4,5-ene.

The manuscript is a significant addition to the literature in chemoenzymatic synthesis of complex molecules and should be published in Nature Comm following revisions.

Revisions

1) Provide experimental outlining the scale up used for the synthesis of the 5 g reported in the manuscript. The current experimental only describes 25mg scale whole cell biocatalytic production

of C19- cortexolone, which is insufficient for the scale of chemistry described in the manuscript. In addition, the experimental does not describe the purification protocol. Thus as is the experimental protocols do not enable reproduction of the work and therefore do not meet the bar for publication.

2) Line 192/193. There is a compound numbering problem here with 13 and 27. Please correct.

3) Please use the correct number of significant digits for the IC50s reported in scheme 3 and the text.

4) When refereeing to the "C19,C21 diols" it should be "C19,C21 diol" or "C19, C21 alcohols". See lines 214 & 241

Reviewer #2 (Remarks to the Author):

The manuscript titled, "A biocatalytic hydroxylation enabled unified approach to C19-hydroxylated steroids" by Want et al. describes the identification of a P450 known to hydroxylate the C19 position of the steroid core and the elaboration of this C19-hydroxy scaffold to a variety pregnane natural products resulting in the structural revision of sclerosteroid B. This work demonstrates the power of biocatalysis in synthesis; however, this work consists of a number of unsurprising results making it difficult to argue for the publication of this manuscript in Nature Communications.

Comments:

1. The authors successfully identified the P450 capable of C19-hydroxylation of the steroid core in *T. cucumeris*, which is known to perform this hydroxylation. This finding could be significant if it translated to effective heterologous expression for fundamental studies (binding, kinetics, structure) on the enzyme; however, this was not possible and the authors reverted to a biotransformation with *T. cucumeris*. While the fermentation conditions could be optimized, this works does not allow for a greater functional or structural understanding of the P450 performing the target hydroxylation.

2. C19-hydroxylated compound 2 was then transformed through a number of standard chemical reactions to pregnane natural products. If the innovative chemistry lies in getting to 2, then better context should be provided for the advantages of accessing 2 biocatalytically. Does existing chemistry allow for the same functionalization on the target substrate? This is discussed generally, but not specifically in the manuscript. Building a stronger case for how this transformation streamlines the synthesis would improve the impact of this work.

3. The advantage the biocatalytic approach potentially provides is the ability to do selective transformation on complex scaffolds. It would improve the impact of this work to understand more on the substrate scope of the P450 capable for this hydroxylation beyond the limited set of highly related compounds tested.

Reviewer #3 (Remarks to the Author):

The authors reported an establishment of an efficient biotransformation method to prepare C19-hydroxylated steroids, which were then used for further synthetic efforts to prepare valuable steroids for medicinal purposes. Based on a reported paper, authors focused on a fungus, *Thanatephorus cucumeris*, a culture of which was shown to catalyze C19 hydroxylation of cortexolone in the past. Due to a low rate of conversion in the original report, authors studied to improve the conversion rate. A P450 gene TcP450-1 responsible for C19 hydroxylation was cloned and heterologously expressed in *Aspergillus oryzae*, however, its conversion rate was not significantly improved. An addition of ferrous ion into a *T. cucumeris* culture greatly improved the conversion of cortexolone into C19-OH product. Finally, using 17-acetyl-cortexolone as a substrate significantly improved the conversion rate (up to 80%) by suppressing the formation of side products. Thus, 17-acetyl group of cortexolone was shown to improve the hydroxylation specificity for C19. These results demonstrated a significant improvement from the

previous literature to prepare C19-OH steroids, and their solution for improving the transformation efficiency is quite original.

In order to enable following up their studies, a little more information is needed in the Methods section.

Line 349, 400, and 423; Please specify *T. cucumeris* was inoculated into what liquid medium? Perhaps, PDA?

Line 351, 387, 403, 414, and 426; Please explain how the substrate cortexolone and its derivatives were added into a liquid media. Are they inoculated as a powder? Or if in a solution, please specify what solvent was used to dissolve the steroids and inoculated (both composition and concentration). If in a case of a powder, are they sterilized before inoculation?

Line 426; Also, for addition of metal ion, please specify the reagent (as a source of ion), amount and how they are inoculated.

Line 296; "poor substrate specificity" should be "poor reaction specificity".

We have revised the manuscript based on the comments and suggestions by the three referees. The point-by-point responses are listed below, and the detailed changes can be found in the marked versions of the manuscript. (Please take note that all the descriptive, positive comments of the referees are omitted, and only referees' comments expressing their concerns/suggestions are listed below, which are followed by our response).

Reply to Referee 1

Critique 1: "Provide experimental outlining the scale up used for the synthesis of the 5 g reported in the manuscript. The current experimental only describes 25 mg scale whole cell biocatalytic production of C19- cortexolone, which is insufficient for the scale of chemistry described in the manuscript. In addition, the experimental does not describe the purification protocol. Thus as is the experimental protocols do not enable reproduction of the work and therefore do not meet the bar for publication."

Response: Thank you very much for this suggestion. We have added experimental procedure for synthesis of 5 g 19-OH cortexolone and described the purification protocol in the revised supplementary information (page S11).

Critique 2: "Line 192/193. There is a compound numbering problem here with 13 and 27. Please correct"

Response: This error has been corrected in the revised manuscript.

Critique 3: "Please use the correct number of significant digits for the IC50s reported in scheme 3 and the text"

Response: This problem has been corrected in the revised manuscript.

Critique 4: "When refereeing to the "C19,C21 diols" it should be "C19,C21 diol" or "C19, C21 alcohols". See lines 214 & 241"

Response: Thank you, this problem has been corrected in the revised manuscript.

Reply to Referee 2:

Critique 1: The authors successfully identified the P450 capable of C19-hydroxylation of the steroid core in *T. cucumeris*, which is known to perform this hydroxylation. This finding could be significant if it translated to effective heterologous expression for fundamental studies (binding, kinetics, structure) on the enzyme; however, this was not possible and the authors reverted to a biotransformation with *T. cucumeris*. While the fermentation conditions could be optimized, this works does not allow for a greater functional or structural understanding of the P450 performing the target hydroxylation.

Response: Thank you very much for the suggestion. You are right, it would be better to find a good heterologous system, especially for those having clean background and easy genetic manipulation system, therefore the C19 hydroxylation activity can be enhanced by overexpression of the P450 enzyme. However, this endeavor usually is challenging as many fungal strains have strong basal activities, e.g. the C-20 reduction activity in the *Aspergillus oryzae*. Therefore, genetic engineering to remove corresponding modification genes must be performed in order to avoid any basal activities; and these efforts are usually time-consuming. Fortunately, we found that the original system *T. cucumeris* is indeed quite good as it has a clean background (no basal activity), fast growth rate and strong C19-hydroxylation activity (after media optimization). Therefore, we didn't try to overexpress the P450 gene in this wild-type strain. But overexpression of the P450 enzyme in the *T. cucumeris* is applicable if we want to get a stronger C19-hydroxylation activity.

Although the heterologous host *Aspergillus oryzae* is not good for biotransformation due to the basal activity, this system is still suitable for protein overexpression. By adding his-tags at the terminal ends of this gene, this P450 enzyme can be purified from the *Aspergillus oryzae* using affinity chromatography (for instance the Ni-ATA resin). So our work should be still helpful for further functional or structural understanding of the P450 performing the target hydroxylation.

Critique 2: C19-hydroxylated compound 2 was then transformed through a number of standard chemical reactions to pregnane natural products. If the innovative chemistry

lies in getting to **2**, then better context should be provided for the advantages of accessing **2** biocatalytically. Does existing chemistry allow for the same functionalization on the target substrate? This is discussed generally, but not specifically in the manuscript. Building a stronger case for how this transformation streamlines the synthesis would improve the impact of this work.

Response: We are grateful to you for giving this good suggestion. Indeed there's only one chemical route to access compound **2** has been reported previously (Please see the Supplementary Scheme 1 for the details), however it requires 13 steps from cortisone acetate to synthesize **2** and suffers from a very low overall yield (2%). In the Line 199, there is a sentence has been added to reinforce the impact of our work by comparing it to the state-of-the-art chemical route.

Critique 3: The advantage the biocatalytic approach potentially provides is the ability to do selective transformation on complex scaffolds. It would improve the impact of this work to understand more on the substrate scope of the P450 capable for this hydroxylation beyond the limited set of highly related compounds tested.

Response: Thank you very much for giving the good suggestion. We indeed have screened a few steroid substrates for biotransformation, including androstenedione and 1,4-androstenedione. We found that the C20-C21 side-chain as well as the C17 hydroxy group is pivotal for this biocatalytic transformation. Part of the results are listed in Table 1 and Figure 2. A more comprehensive study regarding this issue is currently in progress and will be reported in due course.

Reply to Referee 3

Critique 1: "Line 349, 400, and 423; Please specify *T. cucumeris* was inoculated into what liquid medium? Perhaps, PDA?"

Response: Thank you very much. We have corrected it as "YD liquid medium" that means 25 g/L dextrose and 20 g/L yeast extract (pls see "General materials and methods" in the supplementary information).

Critique 2: "Line 351, 387, 403, 414, and 426; Please explain how the substrate

cortexolone and its derivatives were added into a liquid media. Are they inoculated as a powder? Or if in a solution, please specify what solvent was used to dissolve the steroids and inoculated (both composition and concentration). If in a case of a powder, are they sterilized before inoculation?"

Response: Thank very much for pointing out this issue. We have corrected it as "Cortexolone (50 mg, final concentration 0.25 g/L or 0.5 g/L) was dissolved in DMF (0.5 mL) and filtered for sterilization, then added to the medium"

Critique 3: "Line 426; Also, for addition of metal ion, please specify the reagent (as a source of ion), amount and how they are inoculated."

Response: Thank very much. We have corrected it as "Cortexolone (25 mg dissolved in 0.5 mL DMF, final concentration 0.25 g/L) and different kinds of metal ions aqueous solutions (MgSO_4 , ZnSO_4 , MnCl_2 and FeSO_4 , final concentration 1.0 mM/L) were filtered for sterilization, then added to the medium."

Critique 4: Line 296; "poor substrate specificity" should be "poor reaction specificity"

Response: This has been corrected in the revised manuscript.

REVIEWERS' COMMENTS:

Reviewer #1 (Remarks to the Author):

The revised manuscript entitled "A Biocatalytic Hydroxylation Enabled Unified Approach to C19-hydroxylated Steroids" by Wang et al addresses the concerns outlined in my initial review. This manuscript is appropriate for publication in Nature Communications.

Reviewer #2 (Remarks to the Author):

This manuscript details a significant body of work; however, this collection of results are not surprising based on the current literature. Through the review process, the authors have made improvements to the presentation of this work; however, the content was not revised to increase the impact of the results disclosed.

Reviewer #3 (Remarks to the Author):

The re-submitted manuscript has revised all the points raised by the referees. The paper contains useful information for the readers and should be able to reproduce the work according to the methods described. One further point of revision is that in the Methods section (line 378 and line 390) and throughout the Supplementary Information, the unit "mM/L" does not make sense. It should be "mM".

REVIEWERS' COMMENTS

Reviewer #3 (Remarks to the Author):

One further point of revision is that in the Methods section (line 378 and line 390) and throughout the Supplementary Information, the unit “mM/L” does not make sense. It should be “mM”.

Response : This has been corrected.